# Stable Isotope Ratio Analysis for the Authentication of Natural Antioxidant Curcuminoids from *Curcuma longa* (Turmeric)

**DOI:** 10.3390/antiox12020498

**Published:** 2023-02-16

**Authors:** Matteo Perini, Silvia Pianezze, Luca Ziller, Roberto Larcher, Roberto Pace

**Affiliations:** 1Fondazione Edmund Mach, Via Edmund Mach 1, 38098 San Michele all’Adige, Trento, Italy; 2Indena S.p.A., Via Don Minzoni 6, 20049 Settala, Milan, Italy

**Keywords:** curcuminoids, authentication, synthetic, *Curcuma longa*, stable isotope ratio analysis

## Abstract

Curcuminoid complex, a mixture of curcumin, demethoxycurcumin and didemethoxycurcumin, is one of the most popular antioxidants of natural origin, and it has a multitude of other health benefits. It is threatened by the proliferation of counterfeit products on the market containing synthetic curcuminoids whose addition is difficult to identify as they present the three curcuminoid forms in the correct ratios. Consequently, the necessity to detect this fraudulent practice is escalating. Carbon-14 analysis is the most effective available method, but it is also expensive and difficult to implement. This paper describes the first attempt to characterize natural curcuminoids and their synthetic form, used as an adulterant, through the analysis of stable isotope ratios of carbon and hydrogen (expressed as *δ*^13^C and *δ*^2^H). Carbon values greater than −28.6‰ and hydrogen values greater than −71‰ may indicate the addition of synthetic curcuminoids to the natural ones.

## 1. Introduction

Curcuminoids are natural polyphenolic compounds isolated from turmeric roots (*Curcuma longa* and other *Curcuma* species (spp.) [1]), a perennial member of the Zingiberaceae (Ginger) family. They are cultivated primarily in India and in parts of Southeast Asia and are used as a medical herb due to their antioxidant, anti-inflammatory, antimutagenic, antimicrobial, and anti-cancer properties [1]. While curcuminoids are typically present in turmeric in concentrations between 1% and 6% [2], they comprise more than 90% of the weight (*w*/*w*) in the purified extract. The extract, known as “Curcuminoids” or simply “Curcumin,” is a mixture of curcumin (1,7-bis [4-hydroxy-3-methoxyphenyl]-1,6-heptadiene-3,5-dione) [1], also called diferuloylmethane (about 75–80% *w*/*w*), demethoxycurcumin (about 15–20% *w*/*w*), and didemethoxycurcumin, also known as bisdemethoxycurcumin (below 5% *w*/*w*) [3,4]. Curcuminoids have been the top-selling antioxidant herbal item in the U.S. natural and health food channel since 2013 [5]. Due to its chemical structure [6], there is a considerable amount of consumer interest in its potential as a powerful antioxidant that can neutralize free radicals, promote heart, brain, and cognitive health, enhance digestive and liver function, and boost physical performance and mood [7,8,9]. The increase in demand, which is projected to reach $94.3 million by 2022, and the shortage of available ingredients, which has been influenced not only by demand but also by occasional poor harvests and the variable quality of turmeric roots, have led to the adulteration of the curcuminoids.

As reported by Girme et al., the natural curcuminoid extract is increasingly purposely blended with the cheaper synthetic curcumin to obtain a product that is less expensive to produce [10]. This adulterated curcuminoid extract has not been tested for safety and pharmacological activity, and curcumin-containing dietary supplements have recently been blamed for causing acute hepatitis [11], probably due to the use of synthetic curcumin and other toxic food contaminants [7,8,9]. Due to safety concerns, the United States Food and Drug Administration (FDA) has denied the “Generally Recognized As Safe (GRAS)” designation for synthetic curcumin [12].

Two methods have recently been proposed to identify the fraudulent addition of synthetic curcumin to the natural extract. The first is based on the high-performance thin-layer chromatography (HPTLC) and high-performance liquid chromatography-photodiode array (HPLC-PDA) quantification of CIMP-1 (1E, 4Z)-5-hydroxy-1-(4-hydroxy-3-methoxyphenyl) hexa-1,4-dien-3-one, a key intermediate in the synthesis of curcumin of synthetic origin [10]. The second is the identification of the by-product of the chemical synthesis of boron (^10^B), using inductively coupled plasma mass spectrometry (ICP-MS) [10]. However, these two approaches are highly dependent on the type of synthetic pathway used to produce synthetic curcumin.

The United States Pharmacopeia lists specific ratios between the concentrations of curcumin, demethoxycurcumin, and didemethoxycurcumin as a means for controlling curcuminoids [13]. Theoretically, the detection of the last two mentioned forms by liquid chromatography (LC) in turmeric extract can differentiate the natural product from the synthetic one, which lacks both compounds [14]. This type of test is easily rendered useless by a new sophisticated technique: the fraudulent dilution [13] of a natural curcuminoid extract with synthetic curcuminoids, containing the three forms in the right ratios, to obtain spiked curcuminoids. Since the onset of this issue, some analytical approaches to identify adulterated products have been proposed. The most popular and robust method is ^14^C radiocarbon dating, which is expensive, not easily accessible and provided by a limited number of laboratories [15]. The dating method can detect the presence of synthetic curcumin or curcuminoids by identifying the carbon isotopes derived from raw materials of petrochemical derivatives used in the synthetic ingredient.

In recent years, stable isotope ratio analysis (SIRA) found widespread use in food science [16], not only to trace the geographical origin of products, but also to guarantee their authenticity, differentiating between natural and synthetic or biosynthetic ingredients [17,18,19,20,21]. In the latter case, applications of the SIRA involved nutraceutical products or products with pharmacological properties that are increasingly in demand due to their natural origin [18,21,22]. The SIRA is based on the measurement of the ratio between the heavy and light isotopes of the elements carbon (^13^C/^12^C), nitrogen (^15^N/^14^N), sulfur (^34^S/^32^S), oxygen (^18^O/^16^O) and hydrogen (^2^H/^1^H). Chemically identical molecules may have completely different isotope ratios, that vary based on several factors. The carbon stable isotope ratio (*δ*^13^C) is influenced by the C_3_, C_4_ or CAM photosynthetic cycle of the plant from which the molecule derives, or by the fossil source used as a precursor for the chemical synthesis of the molecule. On one hand, as reported by O’Leary at al., while C_4_ plants have *δ*^13^C values between −14 and −12‰, C_3_ plants range from −29 to −25‰ [23]. On the other hand, fossil fuels, which are the primary source in the chemical synthesis of various molecules, have very low *δ*^13^C ranging from −42.5‰ to −25.5‰ [24]. Factors such as the isotopic fractionation, occurring during chemical synthesis, can instead explain the behavior of hydrogen isotopic ratio (*δ*^2^H), which has significantly more positive values in non-natural molecules (e.g., average +63‰ in synthetic vanillin) than in natural ones (e.g., average −32‰ in natural vanillin) [25].

This study investigates, for the first time, the possibility of identifying the fraudulent dilution of natural antioxidant curcuminoids with synthetic curcuminoids by using this powerful analytical approach. Samples of authentic natural curcuminoids, adulterant synthetic curcuminoids, and natural curcuminoids spiked with the synthetic form were analyzed for C and H stable isotope ratios using an isotope ratio mass spectrometer (IRMS) interfaced with an elemental analyzer (EA) and a pyrolyzer (P). Moreover, to ensure the natural and synthetic origin of the curcuminoids used and the percentage of synthetic adulterant added in the spike samples, a ^14^C analysis was performed using a fluid scintillation counter [26].

## 2. Material and Methods

### 2.1. Sampling

Twenty-one samples of authentic (natural origin) curcuminoids were provided by the producer INDENA S.p.A., Milan, Italy. The samples were produced in different years between 2009 and 2017. Six synthetic curcuminoids samples were purchased on the market. The extremely small number of global producers, all located in India, justifies this sample size. In addition, seven curcuminoid complexes spiked with different concentrations of synthetic curcuminoids were produced in the laboratory. The samples were lyophilized and ground into a powder.

### 2.2. Simultaneous Determination of the Three Curcuminoids Components

The three curcuminoid components (curcumin, demethoxycurcumin and didemethoxycurcumin) were measured following the method reported by Sahu et al. [27]. In brief, an HPLC system (Shimadzu, Kyoto, Japan), consisting of two pumps LC 20AD, a photodiode array detector SPD-M20A, with an automatic sample injector, all from Shimadzu, was used. The output signal was monitored and integrated using LC solutions software (Shimadzu, Kyoto, Japan). An Agilent Eclipse XDB-C18 analytical column (4.6 mm × 150 mm, 3.5 mm) was used for chromatographic analysis using a photodiode array (PDA) detection set at 425 nm. A 10 min gradient elution was used with a mixture of 0.1% trifluoroacetic acid and 0.1% formic acid (50:50 *v*/*v*) (solvent 1) and acetonitrile (solvent 2). The percentage of solvent 2 at the start of the gradient was 40% and was increased to 60% (6 min) and then again to 40% up to the end of the analytical run. The flow rate was 0.8 mL/min.

A mixture of acetonitrile and water (50:50) was used as a sample diluent. Methanol was used for the preparation of primary stocks solutions (1 mg/mL), accurately weighed curcumin, demethoxycurcumin and didemethoxycurcumin. These were used to prepare a series of calibration standards with appropriate concentrations, working standard (100 mg/mL) and quality control (QC) samples. Calibration curves (six points) were constructed between the concentration range of 10 and 80 mg/mL.

Fifteen mg of each curcuminoid sample under analysis were transferred to a 15 mL Borosil glass centrifuge tube and centrifuged in acetonitrile at 1500 rpm for 30 min. The supernatant layer (10 mL) was used for direct injection into the HPLC. An electronic balance (Mettler Toledo, Albstadt, Germany), a centrifuge (Thermo Scientific, Bremen, Germany) and an ultra sonicator (SONICA, Soltec, Milan, Italy) were used for sample preparation.

The content of curcumin, demethoxycurcumin and didemethoxycurcumin was expressed as a percentage.

### 2.3. ^14^C Analysis

The determination of biobased carbon content by analyzing the ^14^C content of pulverized food material was performed in accordance with DIN EN 16640:2017-05 [28].

Approximately 20 g of the sample material was combusted in a bomb calorimeter. The sample’s organic carbon content was completely oxidized to CO_2_. The CO_2_ was cleaned for additional oxygen and other gaseous components by using cold traps and dissolving in absorber fluid. After scintillator fluid had been added to the pure CO_2_ (cocktail with 6.8 g CO_2_), the sample cocktail was measured in two glass bottles in a liquid scintillation counter (LSC), i.e., Packard Canberra Tri-Carb 2770 TR/SL, as a repeated measure. The measurement time was approximately 1600 min. Each measurement lasted approximately 50 min, and each sample was measured approximately 10 to 20 times. The measurement was calibrated using ^14^C standards of ^14^C-free CO_2_ and CO_2_ of known ^14^C content. The results were determined by the background count rate and counting efficiency. The result is given in pmC, which can be converted to the content of biogenic carbon using the correction factor of 0.985 in accordance with DIN EN 16640:2017-05. The uncertainty of the method (calculated as two standard deviations when analyzing the same sample at least ten times under reproducible conditions) was 5%-modern.

### 2.4. Stable Isotope Analysis

The ^13^C/^12^C ratio was measured using an isotope mass spectrometer (IsoPrime, Isoprime Limited, Manchester, UK) after complete combustion in an elemental analyzer (VARIO CUBE, Isoprime Limited, Manchester, UK). The ^2^H/^1^H ratio was measured using an IRMS (Finnigan DELTA XP, Thermo Scientific, Bremen, Germany) coupled with a pyrolizer (Finningan DELTA TC/EA, high temperature conversion elemental analyzer, Thermo Scientific, Bremen, Germany). The amount introduced into the above-mentioned instruments to analyze the samples was 0.5 and 0.2 mg, respectively.

Based on the IUPAC protocol [29], ^13^C/^12^C and ^2^H^/1^H values are expressed in the delta scale (*δ*‰) against the international V-PDB (Vienna PeeDee Belemnite) standard according to Equation (1):(1)$$\delta_{ref}{(}^{i}E{/}^{j}E,sample)=\left[\frac{R{(}^{i}E{/}^{j}E,\;\;\;sample)}{R{(}^{i}E{/}^{j}E,\;\;\;ref)}\right]-1$$
where *ref* is the international measurement standard, the sample is the analyzed sample, and *^i^E*/*^j^E* is the ratio of heavier to lighter isotopes. The delta values were multiplied by 1000 and expressed in “per mil” (‰) units or, according to the International System of Units (SI), in unit ‘milliurey’ (mUr).

The isotopic value *δ*^13^C was calculated against working in-house standards (caseins), which were themselves calibrated against international reference materials: fuel oil NBS-22 with *δ*^13^C = −30.03‰, sucrose IAEA-CH-6 with *δ*^13^C = −10.45‰ (International Atomic Energy Agency [IAEA], Vienna, Austria), and L-glutamic acid USGS 40 with *δ*^13^C = −26.39‰ (U.S. Geological Survey, Reston, VA, USA) for ^13^C/^12^C.

Before we carried out the analysis of *δ*^2^H, we left the sample to equilibrate with the laboratory air for two days; it was subsequently placed inside a vacuum dryer with P_2_O_5_ for 48 h and finally placed in a zero-blank autosampler to comply with the principle of identical treatment [30]. Keratins CBS (Caribou Hoof Standard, *δ*^2^H = −157 ± 2‰) and KHS (Kudu Horn Standard, *δ*^2^H = −35 ± 1‰) from the U.S. Geological Survey were used to obtain ^2^H/^1^H values through the creation of a linear equation and by adopting a comparative equilibration procedure [30]. We used these two keratinous standards because of the absence of any international organic reference material with a similar matrix to our samples (curcuminoids). The uncertainty of the method (calculated as two standard deviations when analyzing the same sample at least ten times under reproducible conditions) was 0.3‰ for *δ*^13^C and 4.0‰ for *δ*^2^H values.

### 2.5. Statistical Analysis

XLSTAT (XLSTAT, 2017) was used to statistically evaluate the data. The existence of differences was verified by regression analysis with a confidence level of 95%. A one-way ANOVA was performed to determine the spatially significant differences between variables. Tukey’s honest significant difference (HSD) test for unequal sample sizes was implemented to evaluate significant differences due to geographical origin. Probability (*p*) values of less than 0.05 were used to indicate a significance level.

## 3. Results and Discussion

Based on the ^14^C (^14^C-org.) results, all the samples matched the claimed content (natural or synthetic curcuminoids) (Table 1). Given the uncertainty of the analysis, samples with ^14^C values between 0%-modern and 5%-modern were classified as synthetic, while those with values higher than 85%-modern were classified as natural.

All the samples analyzed, regardless of their synthetic or natural origin, had not dissimilar contents in the three forms: curcumin, demethoxycurcumin and didemethoxycurcumin (see Table 1). The quantification of curcuminoid forms was therefore not useful for discriminating against counterfeit products [13].

### 3.1. δ^13^C of Natural and Synthetic Curcuminoid Complex

The *δ*^13^C of the natural curcuminoid complex ranged between −30.7‰ and −29.0‰, with an average value of −29.8‰. This behavior is in line with the matrix botanical origin. Indeed, turmeric (*Curcuma* spp.) belongs to C_3_ plants, whose *δ*^13^C typically ranges from −29 to −25‰ [23]. Regarding the specific plant used, Khatri et al. reported a value of −25.68‰ for *Curcuma domestica* [31], whereas Marchese et al. determined the carbon isotope composition of *Curcuma longa* L. to be −27.77 ± 0.63 [32].

The gap between the typical *δ*^13^C value of *Curcuma* spp. and the natural curcuminoid complex could be explained by the fact that the different compounds have a characteristic isotope composition due to the different metabolic pathways involved in their synthesis [33]. The same behavior has already been observed in other molecules of vegetable origin, such as Monakolin K. In fact, the average value of the product resulting from the extraction from fermented red rice (RYR) is −25‰, while the average value of the Monakolin K produced by fermentation is −30‰ [21].

The results of the *δ*^13^C analysis of synthetic samples are presented in Table 1. The isotopic values of the analyzed synthetic curcuminoids samples ranged from −29‰ to −26‰, with an average value of −28‰. To understand why synthetic curcuminoids have this *δ*^13^C values, it is necessary to comprehend this molecule production technique. Synthetic curcumin and curcuminoids are synthesized starting from petroleum-derived compounds. Currently, many synthetic production routes to curcumin are variations of Pabon’s method (Figure 1), which originally used acetylacetone and vanillin [34]. Acetylacetone is industrially produced from petrochemicals. Natural vanillin is scarce and expensive to produce, but it can also be manufactured synthetically with petrochemicals at a much lower cost.

As reported by various authors [25,35,36], the range of *δ*^13^C values for synthetic vanillin is between −31.4‰ and −29.4‰, which is outside the range of natural vanillin from the *Vanilla* spp. (with a CAM [crassulacean acid metabolism] photosynthesis cycle), while acetylacetone derived from fossils probably exhibits the typical variability range between −32.5‰ and −23.3‰, as reported by Yeh et al. [37]. The isotopic value obtained in synthetic curcuminoids (average −28‰) is, therefore, justifiable based on the isotopic composition of the two aforementioned starting molecules. While some samples of synthetic curcuminoids have *δ*^13^C values (average = −26.5‰) that are significantly higher (*p* < 0.01) than those typical of natural curcuminoids, others have overlapping values (average = −28.8‰). Therefore, *δ*^13^C analysis does not always detect fraudulent additions of this adulterant to natural curcuminoids, but it could serve as a simple and rapid screening method. Furthermore, few laboratories still routinely carry out the *δ*^2^H analysis and it is therefore necessary to provide reference *δ*^2^H intervals in addition to *δ*^13^C ones, to carry out a more accurate data interpretation. Using a 95% probability level, it is possible to identify a threshold value of −28.6‰ for the *δ*^13^C of the natural curcuminoid complex. Higher *δ*^13^C values may indicate the addition of synthetic curcuminoids.

### 3.2. δ^2^H of Natural and Synthetic Curcuminoid Complex

The *δ*^2^H values found in the two sample types (natural and synthetic curcuminoid) are significantly different and can be used to clearly differentiate between the two products. The natural curcuminoid complex has highly negative values ranging from −120‰ to −82‰ (Table 1). The *δ*^2^H measured in the extract is strictly correlated with that of the source plant, whose isotopic composition mirrors that of the groundwater absorbed by the roots. According to Gat et al. [38], the isotopic signature of precipitation (rainfall) influences the hydrogen isotopic composition of groundwater. The Global Network of Isotopes in Precipitation (GNIP) database, administered by the International Atomic Energy Association (IAEA) and the World Meteorological Organization (WMO), is recognized as a useful method to evaluate the correlation between the *δ*^2^H of rainwater and groundwater in the absence of direct measurement [39]. The GNIP database (http://www-naweb.iaea.org/napc/ih/documents/userupdate/Waterloo/, accessed on 1 January 2023) contains monthly weighted average precipitation (*δ*^2^Hp and *δ*^18^Op) from all continents and islands between 1960 and the present. The temperature and the amount of precipitation significantly influence the *δ*^2^H values of the water. Consequently, based on geographical parameters (latitude, altitude and distance from the source of evaporation), the *δ*^2^H can vary in the precipitation from approximately −60 to +1% worldwide [39]. While *δ*^2^H values range from −70 to −38‰ in China, they range from −38‰ to −6‰ in India. This data can explain the low *δ*^2^H value of natural curcuminoids and, in particular, the lower value found in Chinese samples (average −106‰) compared to Indian samples (average −87‰). In order to proceed with the percentage estimate of natural curcuminoids adulterated with synthetic curcuminoids, the evaluation of the geographical origin must be taken into consideration, as discussed in more detail in Section 3.3.

Synthetic curcuminoids had very high and positive *δ*^2^H values, ranging between +42‰ and +58‰. Considering the synthesis reaction illustrated in Section 3.1, it is evident that most of the hydrogens present in the molecule (eight out of ten) can be traced back to the vanillin molecules. As reported by Greule et al. and Perini et al., the biosynthetic pathways employed in the industrial production of this molecule (replacing the extremely time-consuming and expensive extraction from the orchid) result in a product with a specific hydrogen isotopic composition [25,36]. Positive values (ranging between +38‰ and +104‰) are characteristic of synthetic vanillin.

The significant difference (*p* < 0.01) in hydrogen isotopic composition between natural curcuminoids and their adulterant makes this parameter the most promising for detecting the fraudulent addition of synthetic curcuminoids to the natural product.

Using a 95% probability level, it is possible to identify a threshold value of −71‰ for the *δ*^2^H of natural curcuminoids. Higher values may indicate that a synthetic complex has been added to curcuminoids.

### 3.3. Natural Curcuminoids Complex Spiked with Different Concentrations of Synthetic Curcuminoids

The averages of *δ*^2^H (Figure 2) and *δ*^13^C (Figure 3) were used to quantify the percentage of synthetic curcuminoids added to the natural ones. For each parameter, a graph was created based on the standard deviation (multiplied by t-student) and increasing the percentage of the addition of synthetic curcuminoids to natural ones from 0% to 100%. The mean values of the mixture for each isotopic parameter were calculated as the sum of the mean values of the two groups, multiplied by the percentage contribution to the mixture. The standard deviation resulted from the sum of the standard deviation of the two groups multiplied by the percentage of contribution, according to the propagation of error law in the case of the sum of two or more variables. Seven natural curcuminoids added to the synthetic complex in increasing percentage (from 18% to 72%) were used to validate the graph. The values of these seven samples are shown as orange dots in Figure 2 for hydrogen and in Figure 3 for carbon. The nature of the adulterated samples was confirmed by ^14^C analysis. This ^14^C analysis confirmed the amount of added synthetic components derived from fossil sources in the examined sample (Appendix A).

The analysis of the *δ*^2^H was found to be an effective alternative to the analysis of ^14^C for the detection of fraudulent additions of synthetic complex to natural curcuminoids. With additions of synthetic curcuminoids between 20% and 30%, the *δ*^2^H exceeded the range of variability proposed in Section 3.2 for a natural product (Figure 2).

In adulterated samples with low concentrations of synthetic curcuminoids (between 20% and 30%), the *δ*^13^C (Figure 3) does not differ significantly from the estimated limit value in Section 3.1 (see Figure 3). Therefore, the *δ*^13^C alone cannot be considered as an effective parameter for the detection of synthetic curcuminoid additions; it must always be considered together with hydrogen analysis to correctly evaluate the product, by using the limit suggested in Section 3.2.

Finally, in samples containing substantial additions of synthetic curcuminoids (about 70%), both parameters demonstrated their effectiveness in identifying the presence of synthetic ingredients with significant deviations from the proposed limits.

## 4. Conclusions

The protection of natural antioxidant curcuminoids against a recent new type of adulteration consisting in the addition of synthetic curcuminoids requires the development of increasingly sophisticated, yet rapid and cost-effective, analytical techniques. The ^14^C analysis, despite its undeniable effectiveness, requires a large amount of sample (about 20 g), it is time-consuming (hours considering pretreatment and analyses) and it is performed by few accredited laboratories worldwide only. The isotope ratios of hydrogen and, in some cases, carbon, exhibit significantly different ranges of variability between natural curcuminoids and their synthetic adulterant. These outcomes allow for the identification of not only the sample origin (whether natural or synthetic), but also for the fraudulent addition of synthetic products to the natural complex (spiked samples). Consequently, the analysis of hydrogen isotope ratios can be an effective response because it is relatively quick, inexpensive, and routinely performed in laboratories around the world.

## Figures and Tables

**Figure 1 antioxidants-12-00498-f001:**
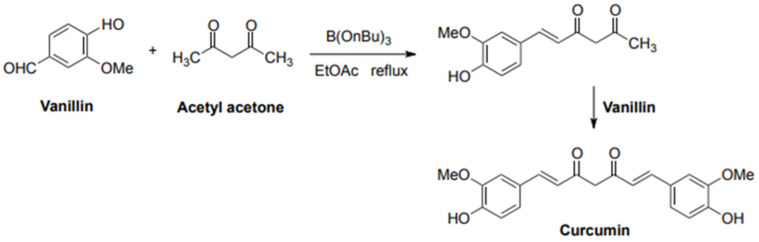
Curcumin synthesis by condensation of acetylacetone and vanillin.

**Figure 2 antioxidants-12-00498-f002:**
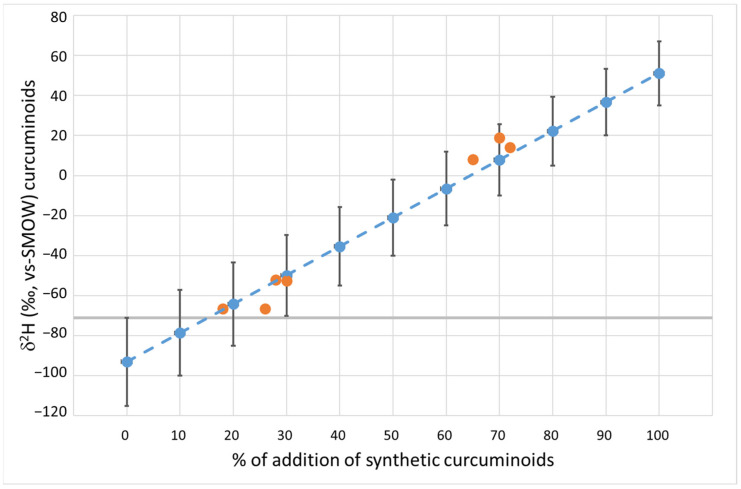
Variations in the δ^2^H values of natural curcuminoids with the addition of synthetic curcuminoids. The grey line defines the 95% threshold limit for authentic natural curcuminoids. Blue dots: mean value. Bars: 95% confidence limit. Orange dots: measured values of prepared mixtures.

**Figure 3 antioxidants-12-00498-f003:**
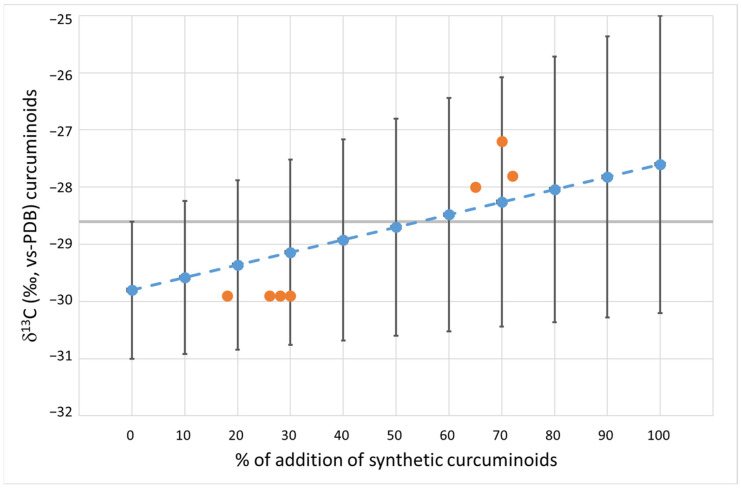
Variations in the *δ*^13^C values of natural curcuminoids with the addition of synthetic curcuminoids. The grey line defines the 95% threshold limit for authentic natural curcuminoids. Blue dots: mean value. Bars: 95% confidence limit. Orange dots: measured values of prepared mixtures.

**Table 1 antioxidants-12-00498-t001:** *δ*^2^H and *δ*^13^C stable isotopic variability, ^14^C content and % of curcumin, demethoxycurcumin and didemethoxycurcumin of natural and synthetic curcuminoids from different geographical origins (India and China). The threshold value 95% for *δ*^2^H and *δ*^13^C of natural curcuminoids are reported.

	Geographical Origin	% Curcumin	% Demethoxy Curcumin	% Didemethoxy Curcumin	Carbon 14 %	*δ*^2^H (‰. vs. V-SMOW)	*δ*^13^C (‰. vs. V-PDB)
Synthetic	India	79.0	17.0	3.9	0.0	52	−26.4
	India	78.0	18.0	3.9	<2	58	−26.3
	India	77.5	18.2	4.0	5.0	47	−26.7
	India	77.6	17.8	3.9	5	42	−29.0
	India	79.0	17.1	3.8	5	44	−29.0
	India	78.0	18.1	3.7	<2	62	−28.4
					Mean	51	−27.6
					SD	8	1.3
Natural	China	78.7	11.2	1.4	90.0	−108	−30.2
	India	76.2	13.8	2.2	96.2	−86	−30.7
	China	78.7	11.2	1.4	96.3	−108	−30.2
	India	71.4	16.4	2.9	97.2	−83	−29.3
	China	79.2	11.3	1.3	85.0	−98	−30.1
	India	71.4	16.4	2.9	97.2	−86	−29.4
	India	74.1	14.1	1.9	97.4	−82	−29.1
	India	74.2	14.4	2.7	97.9	−92	−29.8
	India	76.1	16.9	3.0	98.7	−97	−29.5
	China	71.7	15.7	7.2	99.7	−95	−30.6
	China	78.4	10.9	0.9	99.8	−120	−30.5
	India	78.5	14.6	1.3	100.3	−86	−29.0
	India	78.3	11.4	1.3	97.4	−83	−30.4
	India	76.4	12.4	2.3	97	−84	−30.4
	China	79.0	16.3	2.6	100.7	−110	−30.9
	India	72.0	16.0	2.8	97.8	−83	−29.4
	India	73.1	15.0	2.8	98.6	−87	−29.6
	India	75.6	12.0	2.6	99.7	−90.2	−28.9
	China	76.6	18.0	2.0	97.8	−109.8	−29.7
	India	78.9	15.0	1.9	98.3	−87.5	−29.1
	India	74.3	18.0	2.0	97.4	−87.1	−29.4
					Mean	−93	−29.8
					SD	11	0.6
					Limit 95%	−71	−28.6

## Data Availability

Data is contained within the article and Appendix A.

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
