# Peer review of "Stable Isotope Ratio Analysis for the Authentication of Natural Antioxidant Curcuminoids from Curcuma longa (Turmeric)"

_antioxidants, 2023, doi:10.3390/antiox12020498_

Round 1

Reviewer 1 Report

This is a relatively small but well-constructed and well-presented study.  Not only is the authenticity of cucuminoids important from a consumer/ fraud point of view, but the authors make very good arguments about the potential health risks associated with fraud in this product. 

Some improvements could be made to the methods/ technical part of the manuscript.  Why was d18O not determined?  As demonstrated in this manuscript, carbon is not a good differentiator here and oxygen offers considerable advantages.  Similar to H the O values vary geographically (rainfall as discussed) but also with evaporation cycles and which decouples from H to some extent.  The ratio of the H and O isotope values provides and additional measure to differentiate origins, including geographical and natural vs synthetic.

The measurement of d2H on organic products is complicated by the presence of exchangeable hydrogen.  This leads to part of the signal being derived from the laboratory conditions rather than it origin.  In curcuminoids, there are a few that could exchange, but it is likely this will be  a very small amount.  A greater effect is likely from adhering water. Most protocols require 4 days vacuum drying prior to analysis, and the use of a zero-blank autosampler to comply with the principle of identical treatment.  The methods section does not mention either of these effects or how they were addressed.  This should be discussed.

Was the H determined using pyrolysis (glassy carbon) or reduction (chrome) in the reactor?

The spiking experiments were well conducted and explained.  Perhaps better context could be given by outlining what level of adulteration would be considered to be economically viable.    

What do the error bars represent in the figures? add explanation to the caption. Also need to say in the caption what the blue and orange dots represent.

The conclusions could also make the points that the radiocarbon test require a lot more material (20 g vs a few mg for the stable isotopes).  Radiocarbon also takes a lot longer to collect the data.

It would also be good to quantify the performance difference i.e. what level of adulteration can be reliably detected using radiocarbon vs hydrogen isotopes?

Minor comments

Line 62  sophistication

Line 66 remove 14C

Line 96 ground

Line 116  Isoprime is part of the Elementar UK company

Line 232  added

Line 233 one

Line 270 and 274  deuterium is an isotope of hydrogen, you are measuring hydrogen isotopes.

Author Response

Reviewer 1

This is a relatively small but well-constructed and well-presented study.  Not only is the authenticity of cucuminoids important from a consumer/ fraud point of view, but the authors make very good arguments about the potential health risks associated with fraud in this product. 

Some improvements could be made to the methods/ technical part of the manuscript.  Why was d18O not determined?  As demonstrated in this manuscript, carbon is not a good differentiator here and oxygen offers considerable advantages.  Similar to H the O values vary geographically (rainfall as discussed) but also with evaporation cycles and which decouples from H to some extent.  The ratio of the H and O isotope values provides and additional measure to differentiate origins, including geographical and natural vs synthetic.

The oxygen analysis was also performed on these samples. The results obtained do not provide as clear a discriminating ability as hydrogen. The natural samples vary around +10‰ with respect to the CBS and KHS (USGS standard) while the synthetic ones present two groups with either much higher or much lower values. This also creates problems in the construction of the graph with the cuts as everything will depend on the value of the synthetic curcuminoid used. For this reason it was preferred not to include the data obtained in this work.

The measurement of d2H on organic products is complicated by the presence of exchangeable hydrogen.  This leads to part of the signal being derived from the laboratory conditions rather than it origin.  In curcuminoids, there are a few that could exchange, but it is likely this will be  a very small amount.  A greater effect is likely from adhering water. Most protocols require 4 days vacuum drying prior to analysis, and the use of a zero-blank autosampler to comply with the principle of identical treatment.  The methods section does not mention either of these effects or how they were addressed.  This should be discussed.

We totally agree with the reviewer. The materials and methods part has been corrected by specifying the equilibration times. The laboratory routinely performs hydrogen analyzes on organic matrices by applying a strict sample pre-treatment protocol.

Was the H determined using pyrolysis (glassy carbon) or reduction (chrome) in the reactor?

We used pyrolysis (glassy carbon). The sample does not contain nitrogen.

The spiking experiments were well conducted and explained.  Perhaps better context could be given by outlining what level of adulteration would be considered to be economically viable.    

This figure is highly variable and depends on market conditions and the price of the natural product which is sometimes close to that of the synthetic and sometimes becomes tens of times higher.

What do the error bars represent in the figures? add explanation to the caption. Also need to say in the caption what the blue and orange dots represent.

We corrected the caption

The conclusions could also make the points that the radiocarbon test require a lot more material (20 g vs a few mg for the stable isotopes).  Radiocarbon also takes a lot longer to collect the data.

We corrected as suggested.

It would also be good to quantify the performance difference i.e. what level of adulteration can be reliably detected using radiocarbon vs hydrogen isotopes?

We thank the reviewer you for this comment but this study is not a comparison of methods. The 14C analysis was performed only to guarantee the natural or synthetic or mixed origin of the samples under study.

Minor comments

Line 62  sophistication

Line 66 remove 14C

Line 96 ground

Line 116  Isoprime is part of the Elementar UK company

Line 232  added

Line 233 one

Line 270 and 274  deuterium is an isotope of hydrogen, you are measuring hydrogen isotopes.

We corrected all as request.

Reviewer 2 Report

Abstract is informative and describes novelty and most important results on the stable isotope ratios of δ13C and δ2H. Carbon and deuterium shift values were proposed for adelturation of curcuminoids in the natural extract.

Introduction

The importance of curcuminoid analyses for food industry and consumers is discussed. State of research on the analytical methods and literature is provided. The isotope ratio analysis (SIRA) was described in applications for identification the fradulent dilution of natural curcuminoids with synthetic one. The content of synthetic added  curcuminoids was studied by  14C analysis with fluid scintillation counter. The aim of study is properly formulated

Materials and methods

Authors study samples form the suppliers and provided by food laboratory.

The applied methodology is not new, but acceptable for the purpose of research. Instrumentation is well described.

Statistical Analysis was performed with a XLSTAT (XLSTAT, 2017), one-way 139 ANOVA  and Tukey’s honest significant difference (HSD) test for unequal sample sizes is sufficient.

Results discussion is informative and based on the data and statistical methods used for analysis.

Conclusions are acceptable and based on the results discussion and obtained data.

Paper is worth study and can be useful in  adelturation study in food.

Author Response

Reviewer 2

Abstract is informative and describes novelty and most important results on the stable isotope ratios of δ13C and δ2H. Carbon and deuterium shift values were proposed for adulteration of curcuminoids in the natural extract.

Introduction

The importance of curcuminoid analyses for food industry and consumers is discussed. State of research on the analytical methods and literature is provided. The isotope ratio analysis (SIRA) was described in applications for identification the fradulent dilution of natural curcuminoids with synthetic one. The content of synthetic added curcuminoids was studied by  14C analysis with fluid scintillation counter. The aim of study is properly formulated

Materials and methods

Authors study samples from the suppliers and provided by food laboratory.

The applied methodology is not new, but acceptable for the purpose of research. Instrumentation is well described.

Statistical Analysis was performed with a XLSTAT (XLSTAT, 2017), one-way 139 ANOVA  and Tukey’s honest significant difference (HSD) test for unequal sample sizes is sufficient.

Results discussion is informative and based on the data and statistical methods used for analysis.

Conclusions are acceptable and based on the results discussion and obtained data.

Paper is worth study and can be useful in adulteration study in food.

We thank the reviewer for his positive comments acknowledging the validity of this study.

Reviewer 3 Report

Article

Stable isotope ratio analysis for the authentication of antioxidant curcuminoids from Curcuma longa (Turmeric)

A brief summary

The ubiquity of more or less adulterated products makes credible methods for detecting counterfeits extremely necessary. Therefore, the paper seems to be extremely interesting and valuable. The research is quite well designed, well performed and documented but needs some fixes and additions. Nevertheless, the article is clearly written and the presentation style is appropriate for a scientific journal.

Broad comments

1. The authors assumed that the declarations made by the manufacturers of both natural and synthetic curcumin samples, regarding the percentage composition of the various compounds are true (line 93, Table 1.). They have not done their own research in this aspect. Is it possible to fully trust the manufacturers, and base scientific research on this belief?

2. Data presented showed that synthetic curcumin contained three components. Were there no others? Is the synthetic curcumin a 'blend' with a composition intended to intentionally replicate that of the natural one in the same percentages? Is this composition based on assumptions of similarity to the 'real' product, or is it a coincidence? The authors report a method of producing only one main compound, curcumin (line 177). Are the other 'metabolites' also produced in parallel during industrial synthesis?

Author Response

Reviewer 3

A brief summary

The ubiquity of more or less adulterated products makes credible methods for detecting counterfeits extremely necessary. Therefore, the paper seems to be extremely interesting and valuable. The research is quite well designed, well performed and documented but needs some fixes and additions. Nevertheless, the article is clearly written and the presentation style is appropriate for a scientific journal.

Broad comments

  1. The authors assumed that the declarations made by the manufacturers of both natural and synthetic curcumin samples, regarding the percentage composition of the various compounds are true (line 93, Table 1.). They have not done their own research in this aspect. Is it possible to fully trust the manufacturers, and base scientific research on this belief?

The authors have full confidence in the partner company with which the study was conducted. For years and on various sources, Indena has been doing its utmost to develop methods capable of guaranteeing the natural origin of their products which are often subject to unfair competition by dishonest producers. Our laboratory did not deem it necessary to recalculate the percentages of the three forms.

  1. Data presented showed that synthetic curcumin contained three components. Were there no others? Is the synthetic curcumin a 'blend' with a composition intended to intentionally replicate that of the natural one in the same percentages? Is this composition based on assumptions of similarity to the 'real' product, or is it a coincidence? The authors report a method of producing only one main compound, curcumin (line 177). Are the other 'metabolites' also produced in parallel during industrial synthesis?

Despite the searches we have not found in the literature the exact biosynthetic pathway used to produce the synthetic curcuminoid with the three forms in the right concentrations. We know from unofficial sources that the synthesis is unique and leads directly to the three forms. The synthetic curcuminoid is therefore not produced by adding curcumin alone to a base but is synthesized in toto with the three forms present.

Reviewer 4 Report

This is a very interesting work dealing with the authentication of natural curcuminoid antioxidants by employing quick, inexpensive and routinely methodologies based on stable isotope ratio analysis. The manuscript is very well written, the results are accurate, well described, and the discussion is well supported by literature. In my opinion this manuscript is very useful for scientific community dealing with authenticity issues, specially those focusing on the use of synthetic antioxidants in natural samples. The manuscript can be accepted for publication after a minor revision.

First, I will suggest to include the word “natural” in the manuscript title: “Stable isotope ratio analysis for the authentication of natural antioxidant curcuminoids from Curcuma longa (Turmeric)”

Lines 257-258: Why both parameters need to be considered in conjunction to guarantee authenticity of natural curcuminoid antioxidants? According to your results, this can be addressed by determining only δ2H. In any case, in both parameters are really necessary (which will increase analysis time and price) this need to be better addressed in the manuscript. In the conclusions you clearly state that only δ2H is enough to address the authentication issue.

Minor points:

Line 62: Correct “much more sophisticated sophistication technique” ?

Line 99: Please, provide the corresponding reference for DIN EN 16640:2017-05

Lines 118 and 119: Please, indicate city, country, etc., of Thermo Scientific instruments.

Equation 1: Please, remove (1) from the bottom of the equation.

Maybe Table S1 shown in the Appendix can be included as Table 2 in the Manuscript. I do not see it necessary to be considered a supplementary material.

Author Response

Reviewer 4

This is a very interesting work dealing with the authentication of natural curcuminoid antioxidants by employing quick, inexpensive, and routinely methodologies based on stable isotope ratio analysis. The manuscript is very well written, the results are accurate, well described, and the discussion is well supported by literature. In my opinion this manuscript is very useful for scientific community dealing with authenticity issues, especially those focusing on the use of synthetic antioxidants in natural samples. The manuscript can be accepted for publication after a minor revision.

First, I will suggest including the word “natural” in the manuscript title: “Stable isotope ratio analysis for the authentication of natural antioxidant curcuminoids from Curcuma longa (Turmeric)”

We corrected as request.

Lines 257-258: Why both parameters need to be considered in conjunction to guarantee authenticity of natural curcuminoid antioxidants? According to your results, this can be addressed by determining only δ2H. In any case, in both parameters are really necessary (which will increase analysis time and price) this need to be better addressed in the manuscript. In the conclusions you clearly state that only δ2H is enough to address the authentication issue.

Surely the δ2H analysis is the most promising for identifying the illicit additions of synthetic curcuminoid. Unfortunately many laboratories still do not carry out the analysis of this parameter limiting themselves to that of carbon due to lack of instruments, experience or standards. For this reason, we have decided to include the results of the carbon analysis even if they are not always able to identify counterfeit products. The text has been revised.

Minor points:

Line 62: Correct “much more sophisticated sophistication technique” ?

We rewrote the sentence

Line 99: Please, provide the corresponding reference for DIN EN 16640:2017-05

We inserted the citation as request.

Lines 118 and 119: Please, indicate city, country, etc., of Thermo Scientific instruments.

We corrected as request.

Equation 1: Please, remove (1) from the bottom of the equation.

We corrected as request.

Maybe Table S1 shown in the Appendix can be included as Table 2 in the Manuscript. I do not see it necessary to be considered a supplementary material.

The authors believe that Table 1S provides only more detail than the data already represented in Figures 1 and 2. Inserting it as a Table in the text could be considered a repetition.

Round 2

Reviewer 3 Report

The authors' explanations are persuasive, so they must be believed.